# Effect of Different Cooking Methods on Selenium Content of Fish Commonly Consumed in Thailand

**DOI:** 10.3390/foods11121808

**Published:** 2022-06-19

**Authors:** Alongkote Singhato, Kunchit Judprasong, Piyanut Sridonpai, Nunnapus Laitip, Nattikarn Ornthai, Charun Yafa, Chanika Chimkerd

**Affiliations:** 1Doctor of Philosophy Program in Nutrition, Faculty of Medicine Ramathibodi Hospital and Institute of Nutrition, Mahidol University, Nakhon Pathom 73170, Thailand; alongkote@go.buu.ac.th; 2Institute of Nutrition, Mahidol University, Salaya, Phutthamonthon, Nakhon Pathom 73170, Thailand; piyanut.sri@mahidol.ac.th (P.S.); chanika.chimkerd@gmail.com (C.C.); 3Chemical Metrology and Biometry Department, National Institute of Metrology (Thailand), Pathum Thani 12120, Thailand; nunnapusl@nimt.or.th (N.L.); nattikarn@nimt.or.th (N.O.); charun@nimt.or.th (C.Y.)

**Keywords:** essential trace elements, selenium content, effect of cooking on selenium retention, fish consumption

## Abstract

Although fish are good sources of selenium (Se), an essential trace element for the human body, very limited data exist on Se content in commonly consumed fish in Thailand. Consequently, this study investigated selenium content and the effect of cooking among 10 fish species (5 freshwater and 5 marine) most-commonly consumed by the Thai people. The fish were purchased from three representative wholesale markets within or nearby to Bangkok. All fish species were prepared to determine their edible portions (EP) and moisture contents. Total Se in fresh, boiled, and fried fish were analysed using Inductively Coupled Plasma-Triple Quadrupole-Mass Spectrometry (ICP-QQQ-MS). In general, higher levels of Se were found in marine fish (37.1–198.5 µg/100 g EP in fresh fish, 48.0–154.4 µg/100 g EP in boiled fish, and 52.9–262.4 µg/100 g EP in fried fish) compared to freshwater fish (6.9–29.4 µg/100 g EP in fresh fish, 10.1–26.5 µg/100 g EP in boiled fish, and 13.7–43.8 µg/100 g EP in fried fish). While Longtail tuna showed significantly higher Se content than other fish (*p* < 0.05), boiled Longtail tuna had significantly lower true retention of Se than the other fish (*p* < 0.05). Most fish species retained a high level of selenium (ranged 64.1–100.0% true retention in boiling and frying). Longtail tuna, Short-bodied mackerel, Indo-pacific Spanish mackerel, Nile tilapia, and red Nile tilapia–cooked by boiling and frying–are recommended for consumption as excellent sources of selenium.

## 1. Introduction

Selenium (Se) is an essential trace element necessary for human physiological functions. Long-term inadequate Se intake can lead to health complications and adversely affect quality of life [1]. Most notably, it is associated with an impaired immune system and risk of cancer development. Studies have shown that Se plays a role in downregulating reactive oxygen species (ROS) and T cell differentiation [2,3]. Selenium deficiency-related symptoms include garlic breath, hair loss, spoon nails, an abnormal neurological system, skin and mouth sores, and paralysis [4].

Selenium status is noted as serum Se in the blood with a cut-off concentration of Se < 63 µg/L indicating Se deficiency [5]. For Dietary Reference Intakes (DRIs), the suggested daily Se intake for Thai adults is 55 µg [6]. Multiple studies have reported Se deficiency in many countries. For instance, a cross-sectional study in Spain revealed that 13.9% of children had low serum Se (<60 µg/L) [7]. Another study in Saudi Arabia indicated that 41% of adults had low Se concentration in toenails (< 0.56 µg/g) [8]. In Thailand, it was reported that 56% of children living with HIV had low serum Se [9]. Hence, encouraging adequate Se intake should be advocated among healthy and at-risk groups (e.g., the elderly, children, pregnant and lactating women), and especially among patients with impaired immune systems to support their immunity and lower the risk of malnutrition [10,11].

Thailand is one of the largest fish producing countries with both offshore fishing and farmed fish production. Several species of freshwater and marine fish are commonly consumed by the Thai people as part of Thai traditional food menus. Such fish have a high protein value and, due to abundant production, they are easily accessible at affordable prices in local markets and communities. Previous studies have shown that fish are rich sources of Se. For example, Se concentrations in Japanese Kinmedai, Musu, and Kuromaguro fish were reported to be 1.27, 0.77, and 0.75 mg/kg, respectively [12]. A European study showed that Se concentrations in Atlantic cod, Atlantic halibut, and Atlantic herring were 0.25–0.31, 0.47–0.48, and 0.38–0.61 mg/kg, respectively [13]. A Canadian study noted that local Anchovy, Wahoo, and Flying fish contained Se concentrations of 1.4, 0.8, and 0.5 mg/kg of Se, respectively [14]. In Thailand, however, the database on Se content in fish is limited, especially in terms of commonly consumed cooked fish. The latest report on Se content in Thai fish in 2005 reported that Short-bodied mackerel had 0.88 mg/kg, Walking catfish had 0.47 mg/kg, and Striped snakehead fish had 0.33 mg/kg. However, data on all these species were reported in terms of raw fish and analysed using the spectrofluorometric method [15]. Consequently, this present study aimed to assess total Se content and the effect of different cooking methods among commonly consumed fish in Thailand using inductively coupled plasma triple quadrupole mass spectrometry.

## 2. Materials and Methods 

### 2.1. Chemicals and Reagents

The high purity grade nitric acid (Suprapure 65% HNO_3_) was obtained from Sigma–Aldrich (St Louis, MO, USA). Stock solutions of Se (Se SRM3149), Rh (Rh SRM3144), and certified reference material (CRM) SRM1566b (oyster tissue) were obtained from the National Institute of Standards and Technology (NIST), Gaithersburg, MD, USA. Another CRM was NMIJ7402-a (codfish tissue) obtained from the National Metrology Institute of Japan (NMIJ). Milli-Q^®^ water (Millipore Sigma, MA, USA) was used throughout the study.

### 2.2. Sample Preparation

The 10 most-commonly consumed fish species (5 freshwater and 5 marine species) in Thailand were selected based on food consumption survey data [16] and data in the food composition database [17]. The common names, scientific names, and Thai names of the fish are given in Table 1. All species were purchased from three local fish markets within or nearby to Bangkok (i.e., Tai market, Bangkok Noi market, and Klong Toey market). The 3–4 vendors from each market were randomly selected as representative of total stores. All fish species were prepared, cooked (boiling and frying), homogenized (with skin), freeze-dried, and assessed for moisture content according to procedures described elsewhere [18,19]. Briefly, all fish were weighed before and after removing inedible parts. Deionized water was used for boiling, while palm oil was used for frying. The fish samples used for each cooking method were homogenized, freeze-dried, blended into fine particles, kept in screw-caped plastic bottles, and stored at −20 °C until analysis.

### 2.3. Weight Yield Factor

Weight yield factor (YF) describe the weight change, losses, and gains of water and/or fat, in fish due to cooking. The YF is calculated by the weight of cooked fish (g) divided by that of raw fish (g).

### 2.4. Determination of Se Concentration

Interventionary Inductively Coupled Plasma-Triple Quadrupole-Mass Spectrometry (ICP-QQQ-MS) (Agilent 8800 triple quadrupole ICP-MS, Agilent Technologies, Santa Clara, CA, USA) at the National Institute of Metrology, Thailand, was used to determine total Se content in the fish samples. To remove interferences during the ICP-QQQ-MS measurement, collision and reaction gases modes were applied to detect the several masses of Se with enhanced sensitivity and accuracy. The stock and working standard solutions of Se were prepared by serial dilution with 2% HNO_3_ to create an external calibration curve. To correct any changes in instrument operating conditions, a stock solution of Rh was prepared and added to the final solution as an internal standard.

The procedure used for sample solution preparation was adapted from a previous report [20]. Briefly, 0.5 g of each freeze-dried sample was added to a glass vial followed by 5 mL of Suprapure 65% HNO3. The prepared solutions including blanks, CRMs, and fish samples were digested in triplicate by Anton Paar Multiwave 7000 (Anton Paar, Graz, Austria) under the conditions as shown in Table 2. Once digestion was completed, each digested sample was transferred, rinsed, and made up to volume with deionized water to 40 mL in a polypropylene tube. Thereafter, an aliquot volume was placed into another polypropylene tube and an internal standard (Rh) solution was added. The total volume for each prepared solution (10 mL) was used for Se analysis using ICP-QQQ-MS.

### 2.5. Accuracy and Precision of Analytical Method 

To validate the Se analysis by ICP-QQQ-MS, the CRMs of SRM1566b (oyster tissue) and NMIJ7402-a (codfish tissue) were digested in triplicate and analysed for Se content using ICP-QQQ-MS. The analysed concentrations of Se were calculated into dry matter as indicated in the certifications of the CRMs using the moisture content in their samples. Finalized concentrations of Se in CRMs were statistically compared to check the accuracy of Se measurement. In this study, the Se concentration of SRM1566b was 2.08 ± 0.13 mg/kg (certified value at 2.06 ± 0.15 mg/kg) and for NMIJ7402-a was 1.93 ± 0.07 mg/kg (certified value at 1.80 ± 0.20 mg/kg). The precision (as relative standard deviation) of CRMs was 6.59% for SRM1566b and 4.00% for NMIJ7402-a, which indicated that the Se analysis method gave reliable results for both accuracy and precision.

### 2.6. Limit of Detection (LOD) and Limit of Quantitation (LOQ) of Analytical Method

The LOD estimation was determined in terms of the lowest concentration of Se in fish samples using the following equation.
$$LOD=\frac{3\times standard\;error}{slope}$$

For LOQ, estimation was determined in terms of the lowest concentration of Se in fish samples using the following equation.
$$LOQ=\frac{10\times standard\;error}{slope}$$

The concentration at the LOQ level must be proven to ensure that analysis at this level can be determined with acceptable accuracy and precision. To prove LOQ, the concentration of Se was prepared by spiking the Se standard in blank solutions (*n* =10) and these solutions were analysed using ICP-QQQ-MS. For this study, the LOQ method was 3 µg/kg with an obtained recovery of 103.1% and a precision of 6.2% RSD, which passed an acceptable criterion for the LOQ [21,22]. Finally, this LOQ was calculated in the fish sample from the following equation.
$$\mathrm{LOQ}\;\mathrm{for}\;\mathrm{solid}\;\mathrm{sample}=LOQ\times \frac{\mathrm{weight}\;\mathrm{of}\;\mathrm{digested}\;\mathrm{sample}\;\mathrm{solution}\;\left(g\right)}{\mathrm{weight}\;\mathrm{of}\;\mathrm{sample}\;\left(g\right)}$$

### 2.7. True Retention of Se in Cooked Fish

To determine the effect of different cooking methods on Se content in cooked fish, all fish species were weighed to three significant digits using an analytical balance and recorded both before and after cooking. The true retentions of Se in cooked fish were calculated from the following equation [23].
$$\%\;\mathrm{True}\;\mathrm{retention}=\frac{µg\;\mathrm{Se}\;\mathrm{per}\;100g\;\mathrm{of}\;\mathrm{cooked}\;\mathrm{fish}\;\times \;\mathrm{weight}\;\mathrm{of}\;\mathrm{cooked}\;\mathrm{fish}}{µg\;\mathrm{Se}\;\mathrm{per}\;100g\;\mathrm{of}\;\mathrm{raw}\;\mathrm{fish}\;\times \;\mathrm{weight}\;\mathrm{of}\;\mathrm{raw}\;\mathrm{fish}}\;\times 100$$

### 2.8. Statistical Analysis

All data including percentages of edible portions, yield factors, loss of moisture, moisture contents, and Se concentrations in fresh, boiled, and fried fish were reported as mean ± standard deviation (SD). The statistical significance of Se contents at *p* < 0.05 for the different types of fish and their true retention after cooking were evaluated and using two-way ANOVA with interaction followed by Tukey’s Honestly Significant Difference to test multiple pairwise comparisons. Statistical analysis was performed using IBM^®^ SPSS Statistics for Windows, Version 21.0.

## 3. Results

### 3.1. Edible Portions, Yield Factors, and Moisture Contents

The edible portion (EP) is defined as the amount of food that is usually eaten. The EPs of fish in this study were calculated by wet weight after removing inedible parts of the fish, such as scales, bones, internal organs, etc. The results revealed that the range of EP in freshwater fish in fresh form (Table 3) was 46–68%, which is less than that of marine fish (52–83%) (Table 4). The yield factor is defined as the retained weight of food after processing or cooking. This study’s results revealed that the yield factors for both boiled freshwater and marine fish stayed within the ranges of 0.8–0.9 for freshwater fish (Table 3) and 0.6–0.8 for marine fish (Table 4). For fried fish, the yield factor of freshwater fish was 0.5–0.8, while for marine fish it was 0.7–0.8. 

The moisture contents were low in fried freshwater fish (40–74 g/100g) (Table 3) and marine fish (54–73 g/100g) (Table 4) compared to fresh and boiled fish. This finding was in line with a high percentage of loss of moisture in fried fish (13.9–25.0% in freshwater fish and 11.8–27.0% in marine fish) compared to boiled fish (1.1–5.9% in freshwater fish and 1.4–6.6% in marine fish). 

### 3.2. Selenium Content

Total Se contents of the fish are shown in Table 3 and Table 4. Results revealed that marine fish contained higher Se content than freshwater fish in all samples except for Pangasius Dory. The range of Se concentration in fresh freshwater fish was 6.9–29.4 µg/100 g EP and in marine fish was 37.1–198.5 µg/100 g EP. Boiled fish contained higher Se content (10.1–26.5 µg/100 g EP in freshwater fish and 48.0–154.4 µg/100 g EP in marine fish) than that of fresh fish, except for Striped snakehead, Longtail tuna, and Indo-pacific Spanish mackerel. Fried fish had much higher Se content (13.7–43.8 µg/100 g EP for freshwater fish and 52.9–262.4 µg/100 g EP for marine fish) than that of fresh fish except for Short-bodied mackerel. 

After determining differences in Se in fish by two-way ANOVA with interaction followed by Turkey’s HSD post hoc test, a significant difference in the combined effects of fish species and cooking methods (Figure 1A) was found (*p* = 0.003). In particular, a significant difference was found in Se content among different types of fish (*p* < 0.05) (Table 5). Longtail tuna showed significantly the highest Se levels (*p* < 0.05, estimated marginal means was 208.4 μg per 100 g EP) compared to other fish (11.3–89.7 μg per 100 g EP). There was also a significant difference in the effect of the cooking method on Se content (*p* < 0.001) (Table 5 and Table 6, estimated marginal means ranged from 46.0 to 65.4 μg per 100 g EP). Fried fish showed significantly highest Se levels (*p* < 0.05, estimated marginal means was 65.4 μg per 100 g EP) compared to fresh and boiled fish (46.0 and 44.8 μg per 100 g EP, respectively). 

### 3.3. Effect of Different Cooking Methods on True Retention of Se

Data on true retention (TR) of Se in the selected freshwater and marine fish are presented in Table 3 and Table 4, respectively. Most freshwater fish had 100 %TR of Se for both boiling and frying, except for the Striped snakehead, which was 70.5 %TR for boiling and 98.2 %TR for frying. For marine fish, results revealed that true retention was in the same range for boiling (64.4–100%) and frying (64.1–100%). 

Results from two-way ANOVA with interaction followed by Turkey’s HSD post hoc test found significant differences in the combined effects of different species of fish and cooking methods on percentage of true retention (*p* < 0.001). Figure 1B shows the combined effects of fish species and cooking methods and indicates that cooking methods affect different magnitudes of percentages of true retention among different fish species. For instance, boiled Striped snakehead (70.5%) and Longtail tuna (64.4%) showed a lower percentage of true retention than for the other fish (Table 3 and Table 4). In addition, true retention of other boiled fish, such as Indo-pacific Spanish mackerel (78.8%) and short-bodied mackerel (88.8%), were lower than that of Nile tilapia, Walking catfish, Common silver barb, Red tilapia, and Giant sea perch. For fried fish, results revealed that Short-bodied mackerel (64.1%) and Giant sea perch (74.1%) had lower percentages of true retention than other fish (Table 4). 

There was a significant difference in both the species of fish and the effect of cooking methods on %TR (*p* < 0.001). Common silver barb, Nile tilapia, Red Nile tilapia, Walking catfish, and Pangasius Dory showed significantly highest levels of %TR (*p* < 0.05, estimated marginal means ranged between 97.4–100 %TR) compared to other fish (76.5–89.4 %TR) (Table 5 and Table 6). For cooking methods, there was also a significant difference in %TR between boiling and frying (Table 5 and Table 6). Fried fish showed significantly highest %TR (*p* < 0.05, estimated marginal means was 95.1 %TR) compared to boiled fish (90.5 %TR). 

## 4. Discussion

### 4.1. Edible Portions, Yield Factors, and Moisture Contents

The range of edible portion (EP) in freshwater fish in fresh form is less than that of marine fish. This finding can be explained by the Thai traditional household practice of removing the scales and internal organs from freshwater fish during preparation, while for some marine fish in this study these parts are not removed. For boiled fish, the EP of boiled freshwater fish (53–79%) was in the same range as marine fish (47–73%). For fried fish, the EP of fried freshwater fish (39–58%) was similar to that of marine fish (41–60%). The ranges of EP in this study, however, were different from those of fish reported for Bangladesh (62–85%) [24]. For yield factor, the ranges in terms of yield factor could be affected by the different compositions of fish tissues, which is one of the factors that affect weight change after cooking [25]. In addition, a previous study pointed out that different fat contents in foods also contribute to this variation [26]. The result on moisture contents was in line with a high percentage of loss of moisture in fried fish compared to boiled fish. The main reason for these results could be that frying involves heat processing at high temperatures which highly contributes to moisture loss in foods, especially fish where the main weight composition is from moisture [27,28,29,30].

### 4.2. Selenium Contents

Most of fried fish had much higher Se content than that of fresh fish. Results revealed that marine fish contained higher Se content than freshwater fish in all samples except for Pangasius Dory. Boiled fish contained higher Se content than that of fresh fish except for Striped snakehead, Longtail tuna, and Indo-pacific Spanish mackerel. Fried fish had much higher Se content than that of fresh fish except for Short-bodied mackerel. These results agree well with a previous study that noted frying showed minimal loss of Se in Se-biofortified cereals compared to other Chinese cooking methods [31]. This present study’s overall findings indicate that Se contents in commonly consumed Thai fish are comparable to other species of fish in Europe that contained Se ranging between 22 to 61 µg/100 g [13], as well as fish in Japan that contained Se 12–127 µg/100 g [12]. A significant difference in the combined effects of fish species and cooking methods was found. In particular, a significant difference was found in Se content among different types of fish (Table 5). Longtail tuna showed significantly the highest Se levels compared to other fish. There was also a significant difference in the effect of the cooking method on Se content (Table 5 and Table 6). Fried fish showed significantly highest Se levels compared to fresh and boiled fish. This finding was not similar to a previous study that investigated the effect of different cooking methods on vitamin D in Thai fish that suggested no effect of cooking methods on vitamin D content [19]. The different chemical forms of Se that are contained in fish may be one of the main factors that affect the loss of Se during cooking. Selenomethionine and selenocyteine are the main forms of Se in most fish [32], while some fish such as tuna contained selenoneine as the main chemical form in their blood [33]. The mentioned Se forms are low molecular weight and easily removed along with the water [34]. Therefore, this could be one of the reasons explaining why boiling fish leads to more lost Se than frying.

### 4.3. Effect of Different Cooking Methods on True Retention of Se

Most freshwater fish had 100% true retention of Se for both boiling and frying. For marine fish. Due to the large size of Striped snakehead, Indo-pacific Spanish mackerel, and Longtail tuna, they were cut into small pieces (about 1–2 inches in diameter) before boiling. Most freshwater fish were different from the marine fish in physical characteristics in that they possessed scales. The scales of fish may minimize heat exposure during cooking [35,36]. These overall findings were in accordance with data established by the Food and Agriculture Organization of the United Nations (FAO) on the true retention of Se obtained in fish after cooking wherein a range of 90–100% showed that Se is good in terms of heat resistance [37]. Results found significant differences in the combined effects of different species of fish and cooking methods on percentage of true retention. The combined effects of fish species and cooking methods and indicates that cooking methods affect different magnitudes of percentages of true retention among different fish species. 

Fried Giant sea perch had lower percentages of true retention than other fish. This may be because these fish are large, and thus they were cut into small pieces (about 1–2 inches in diameter) before cooking, which increases surface area and the effect of heat during cooking. On the other hand, the Short-bodied mackerel was small compared to the other fish species, but it also showed lower %TR (88.8% TR for boiling and 64.1% for frying). This is because this fish has a thin skin, which may cause a high loss in Se during cooking, especially frying where it is in direct contact with the high temperature cooking oil (about 180 °C).

Common silver barb, Nile tilapia, Red Nile tilapia, Walking catfish, and Pangasius Dory showed significantly highest levels of %TR compared to other fish. For cooking methods, there was also a significant difference in %TR between boiling and frying. Fried fish showed significantly highest %TR (95.1 %TR) compared to boiled fish (90.5 %TR). This data on true retention of Se is comparable to data reported for fish in Europe, such as the Gilthead seabream (*Sparus aurata*) that had a Se true retention percentage of 90–100% for steaming [38]. Consequently, the findings show a high true retention of Se in commonly consumed Thai fish.

A previous study reported on heavy metal contamination in several fish species and other seafood [39]. The major contaminating element was mercury (Hg) and it is commonly bound with Se. The data on Se:Hg molar ratio in fish was reported at 0.23–1 [40,41]. Other heavy metals that could contaminate aquatic animals, such as Cadmium (Cd), Arsenic (As), and Lead (Pb) [42], are well known as harmful to the human body and have been reported to elevate the risk of cancer development [43]. Hence, future studies are needed to determine heavy metals concentrations in commonly consumed fish in Thailand to ensure that the amounts of contaminated elements are below the maximum levels for fish and seafood according to the criteria of FAO (i.e., Pb < 0.3 mg/kg, Hg < 1.2–1.6 mg/kg, etc.) [44].

The high bioaccessibility of Se in fish has been reported at 50–80% [45]. This emphasizes the benefits of fish consumption, not only as sources of high protein and unsaturated fatty acids, but also as good sources of bioavailable Se.

Finally, Se can be found in several forms. The organic forms of Se, such as selenomethionine and selenocysteine, are the main forms found in fish and the meat of other animals [46,47]. Inorganic forms of Se, such as selenate and selenite, can be toxic and less efficient in the human body compared to organic forms [48,49]. One limitation of this present study was not performing a Se speciation analysis. Consequently, determination of the different forms of Se and bioaccessibility of Se obtained in these selected fish is suggested for future study.

## 5. Conclusions

This study confirmed a high Se content obtained in commonly consumed fish in Thailand that are cooked using different methods based on Thai traditional household practices. Marine fish had a higher range of Se content compared to freshwater fish, although the percentage of true retention of Se was comparable between freshwater and marine fish. In conclusion, Thai fish are good sources of Se and have high true retention when using several cooking methods. Their increased consumption should be promoted among healthy and at-risk groups to strengthen immune systems and improve overall nutritional status.

## Figures and Tables

**Figure 1 foods-11-01808-f001:**
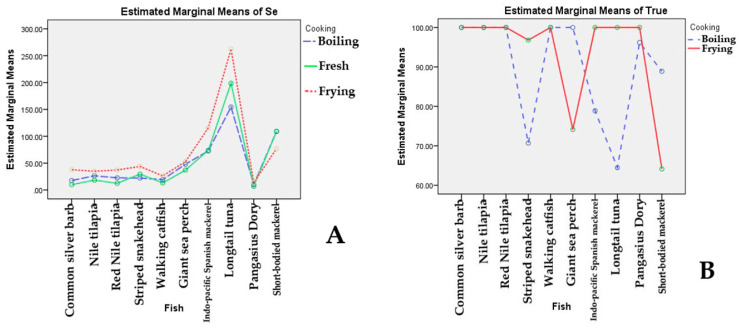
The combined effects of different species of fish and cooking methods on Se concentration (**A**) and on the percentage of true retention (**B**).

**Table 1 foods-11-01808-t001:** The selected top 10 most-commonly consumed fish used in this study.

Common Name	Fish with Scale	Scientific Name	Local Name	Purchase
				(Month/Year)
Fresh water fish:				
Common silver barb	Yes	*Barbonymus gonionotus*	Pla-ta-pian	Sep. 2018
Nile tilapia	Yes	*Oreochromis niloticus*	Pla-nin	Aug. 2018
Red Nile tilapia	Yes	*Oreochromis niloticus-mossambicus*	Pla-tub-tim	Aug. 2018
Striped snakehead	Yes	*Channa striata*	Pla-chon	Aug. 2018
Walking catfish	No	*Clarias macrocephalus*	Pla-duk	Aug. 2018
Marine Fish:				
Giant sea perch	Yes	*Lates calcarifer*	Pla-kha-pong-khaw	Sep. 2018
Indo-pacific Spanish mackerel	Yes	*Scomberomorus guttatus*	Pla-in-see	Sep. 2018
Long tail tuna	No	*Thunnus tonggol*	Pla-O	Dec. 2018
Pangasius Dory	No	*Pangasius hypophthalmus*	Pla-dolly	Nov. 2018
Short-bodied mackerel	No	*Rastrelliger brachysoma*	Pla-tu	Sep. 2018

**Table 2 foods-11-01808-t002:** Parameters used for digestion and analysis of selenium.

Methods	Setting	
Microwave system parameter:		
Estimated sample weight	0.5 g	
Starting pressure	40 bar	
Pressure	160 bar	
Step time	Step 1: 25–220 °C 20 min Step 2: 220 °C 20 min	
Cooling temperature	50 °C	
Pressure release rate	8.0 bar/min	
ICP-QQQ-MS parameter:	He mode	O_2_ mode
RF power	1550 W	
Sampling depth	8 mm	
Carrier gas flow rate	1.05 L min^−1^	
Makeup gas flow rate	0.2 L min^−1^	
	3 mL/min	30%
Monitor masses	^77^Se, ^78^Se, ^82^Se, ^78^Se^16^O^+^, ^80^Se^16^O^+^, ^82^Se^16^O^+^	

**Table 3 foods-11-01808-t003:** Percentage of edible portion, yield factor, and moisture content of three individual sets from each type of freshwater fish, data expressed as mean ± SD (*n* = 3).

Fish Name	Type of Sample	Edible Portion (%)	Yield Factor	Moisture (g/100 g)	Se concentration (µg/100 g of Product)	True Retention of Se (%)
Common silver barb	Fresh (with skin)	50 ± 7	-	74 ± 3.2	9.9 ± 6.8	-
	Boiled (with skin)	56 ± 3	0.8 ± 0.0	71 ± 1.1	17.5 ± 2.9	100.0 ± 0.0
	Fried (with skin)	41 ± 9	0.5 ± 0.0	40 ± 2.7	37.7 ± 10.4	100.0 ± 0.0
Nile tilapia	Fresh (with skin)	46 ± 6	-	76 ± 1.8	18.4 ± 4.2	-
	Boiled (with skin)	53 ± 4	0.9 ± 0.0	73 ± 1.1	26.5 ± 10.9	100.0 ± 0.0
	Fried (with skin)	39 ± 3	0.7 ± 0.0	57 ± 0.6	34.9 ± 18.4	100.0 ± 0.0
Red tilapia	Fresh (with skin)	50 ± 2	-	73 ± 0.4	12.4 ± 6.7	-
	Boiled (with skin)	60 ± 7	0.9 ± 0.0	70 ± 2.3	22.6 ± 4.3	100.0 ± 0.0
	Fried (with skin)	44 ± 5	0.7 ± 0.0	59 ± 3.0	37.2 ± 5.7	100.0 ± 0.0
Striped snakehead	Fresh (with skin)	50 ± 3	-	74 ± 0.4	29.4 ± 11.4	-
	Boiled (with skin)	56 ± 4	0.9 ± 0.0	72 ± 1.3	22.2 ± 3.8	70.5 ± 1.2
	Fried (with skin)	41 ± 3	0.7 ± 0.0	57 ± 1.1	43.8 ± 6.5	98.2 ± 5.8
Walking catfish	Fresh (skinless)	51 ± 6	-	68 ± 0.6	13.1 ± 7.2	-
	Boiled (skinless)	58 ± 2	0.9 ± 0.0	64 ± 2.3	19.4 ± 1.6	100.0 ± 0.0
	Fried (skinless)	41 ± 4	0.8 ± 0.0	54 ± 7.8	26.3 ± 9.3	100.0 ± 0.0

**Table 4 foods-11-01808-t004:** Percentage of edible portion, yield factor, and moisture content of 3three individual sets from each type of marine fish, data expressed as mean ± SD (*n* = 3).

Fish Name	Type of Sample	Edible Portion (%)	Yield Factor	Moisture (g/100 g)	Se concentration (µg/100 g of Product)	True Retention of Se (%)
Giant sea perch	Fresh (with skin)	54 ± 3	-	74 ± 3.0	37.1 ± 8.4	-
	Boiled (with skin)	65 ± 3	0.8 ± 0.2	73 ± 1.8	48.0 ± 21.2	100.0 ± 0.0
	Fried (with skin)	43 ± 3	0.7 ± 0.0	54 ± 2.0	52.9 ± 10.5	74.1 ± 1.7
Indo-pacific Spanish mackerel	Fresh (with skin)	83 ± 5	-	75 ± 0.3	73.7 ± 8.7	-
	Boiled (with skin)	73 ± 0	0.6 ± 0.0	71 ± 2.1	72.6 ± 10.0	78.8 ± 0.0
	Fried (with skin)	60 ± 0	0.8 ± 0.0	58 ± 1.5	115.5 ± 5.1	100.0 ± 0.0
Longtail tuna	Fresh (skinless)	66 ± 2	-	71 ± 2.2	198.5 ± 49.51	-
	Boiled (skinless)	58 ± 1	0.8 ± 0.0	68 ± 2.9	154.4 ± 44.5	64.4 ± 4.9
	Fried (skinless)	52 ± 3	0.8 ± 0.0	61 ± 0.6	262.4 ± 72.9	100.0 ± 0.0
Pangasius Dory	Fresh (skinless)	68 ± 0	-	86 ± 0.7	6.9 ± 2.1	-
	Boiled (skinless)	79 ± 2	0.8 ± 0.2	85 ± 1.5	10.1 ± 0.5	100.0 ± 0.0
	Fried (skinless)	58 ± 0	0.6 ± 0.0	74 ± 1.9	13.7 ± 2.7	100.0 ± 0.0
Short-bodied mackerel	Fresh (skinless)	52 ± 5	-	76 ± 1.8	108.8 ± 14.7	-
	Boiled (skinless)	47 ± 3	0.8 ± 0.0	71 ± 2.4	109.1 ± 28.4	88.8 ± 0.0
	Fried (skinless)	41 ± 3	0.8 ± 0.1	67 ± 4.9	76.9 ± 20.5	64.1 ± 0.0

**Table 5 foods-11-01808-t005:** Estimated marginal means of interaction effect of species of fish and cooking methods on Se content and true retention of Se (calculated from two-way ANOVA) (*n* = 3).

Common Name	Se Content (μg/100 g of Product, Mean ± Standard Error)		True Retention of Se (%, Mean ± Standard Error)	
	Boiled	Fried	Boiled	Fried
Common silver barb	17.6 ± 3.0 ^i,l^	37.8 ± 10.4 ^f,m^	100.0 ±1.5 ^a,k,l^	100.0 ± 1.5 ^a,k^
Nile tilapia	26.5 ± 10.9 ^e,l^	34.9 ± 18.4 ^h,l,m^	100.0 ± 1.2 ^b,k^	100.0 ± 1.2 ^b,k^
Red Nile tilapia	22.6 ± 4.4 ^f,l^	37.3 ± 5.8 ^g,l,m^	100.0 ± 1.5 ^c,k,l^	100.0 ± 1.5 ^c,k^
Striped snakehead	22.2 ± 3.9 ^g,l^	43.8 ± 6.5 ^e,l,m^	70.7 ± 1.2 ^i^	96.8 ± 1.2 ^h,k^
Walking catfish	19.4 ± 1.7 ^h,l^	26.3 ± 9.3 ^i,l,m^	100.0 ± 1.2 ^d,k,l^	100.0 ± 1.2 ^d,k^
Giant sea perch	48.0 ± 21.3 ^d,l^	52.9 ± 10.6 ^d,m^	100.0 ± 1.5 ^e,k,l^	74.1 ± 1.5 ^i^
Indo-pacific Spanish mackerel	72.7 ± 10.1 ^c,k^	115.6 ± 5.1 ^b,k^	78.8 ± 1.5 ^h,l^	100.0 ± 1.5 ^e,k^
Longtail tuna	154.4 ± 44.6 ^a,k^	262.4 ± 72.9 ^a^	64.4 ± 1.5 ^j^	100.0 ± 1.5 ^f,k^
Pangasius Dory	10.1 ± 0.5 ^j,l^	13.7 ± 2.7 ^j,m^	96.2 ± 1.5 ^f,k,l^	100.0 ± 1.5 ^g,k^
Short-bodied mackerel	109.2 ± 28.4 ^b,k^	76.9 ± 20.5 ^c,k,l^	88.8 ± 1.5 ^g,l^	64.1 ± 1.5 ^j^

Estimated marginal means values with the different superscript letters in the same column were significantly different for a given variable (*p* < 0.05 two-way ANOVA followed by Tukey’s HSD post hoc multiple comparisons).

**Table 6 foods-11-01808-t006:** Estimated marginal means of Se concentration and percentage of Se true retention by the main effects of different species of fish and cooking methods (calculated from two-way ANOVA) (*n* = 3).

Common Name	Estimated Marginal Means ± Standard Error	
	Se (μg/100 g of Product)	True Retention (%)
Different species of Fish:		
Common silver barb	27.6 ± 13.2 ^c^	100.0 ± 5.4 ^a^
Nile tilapia	30.7 ± 14.2 ^c^	100.0 ± 4.4 ^a^
Red Nile tilapia	29.9 ± 9.4 ^c^	100.0 ± 5.4 ^a^
Striped snakehead	33.0 ± 12.7 ^c^	83.7 ± 4.4 ^b,c^
Walking catfish	22.8 ± 7.0 ^c^	100.0 ± 4.4 ^a^
Giant sea perch	47.9 ± 14.7 ^c^	87.0 ± 5.4 ^b^
Indo-pacific Spanish mackerel	89.7 ± 28.1 ^b^	89.4 ± 7.6 ^b^
Longtail tuna	208.4 ± 79.4 ^a^	82.2 ± 5.4 ^b,c^
Pangasius Dory	11.3 ± 2.1 ^d^	97.4 ± 6.2 ^a^
Short-bodied mackerel	80.7 ± 3.2 ^b^	76.5 ± 7.6 ^c^
Cooking methods in different species of fish:		
Fresh	46.0 ± 55.7 ^b^	-
Boiling	44.8 ± 44.7 ^c^	90.5 ± 13.9 ^b^
Frying	65.4 ± 66.0 ^a^	95.1 ± 10.8 ^a^

Values with different superscript letters of species of fish or cooking methods in the same column were significantly different for a given variable (*p* < 0.05 two-way ANOVA followed by Tukey’s HSD post hoc multiple comparisons).

## Data Availability

Data is contained within the article.

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
