# Peer review of "Effect of Different Cooking Methods on Selenium Content of Fish Commonly Consumed in Thailand"

_foods, 2022, doi:10.3390/foods11121808_

Round 1
Reviewer 1 Report
In this manuscript entitled " Effect of different cooking methods on selenium content in commonly consumed Thai fish", the authors report selenium content in fish landed in Thailand and after cooking. I think the data presented in this MS is very valuable. I have comments explained below. I hope that my comments are very useful for improving this research.
Comments
(1) Title: I think the English is wrong. How about the title " Effect of different cooking methods on selenium content of fish commonly consumed in Thailand"?
(2) Table 1: Please indicate the year and month the fish was purchased. There may be seasonal variations.
(3) Table 3: Please indicate the units for Edible portion and Yield factor, as they are not shown.
(4) Table 4: Please indicate the units for Edible portion and Yield factor, as they are not shown.
(5) Figure 1: The text in the figures is too small to read, please make the text larger.
(6) Table 5: There are places where the values of mean and standard deviation do not have the same number of digits. Please correct appropriately.
(7) L254-256: Two-way ANOVA is not mentioned in the statistical analysis section of the method. Please describe in the Methods section.
Section 4,2: The difference in the percentage of selenium remaining after cooking in different species of fish may be due to differences in the chemical form of selenium compound contained in the fish. Most of the selenium in fish is contained as selenocysteine in proteins such as glutathione peroxidase. On the other hand, large fish such as tuna contain not only selenocysteine, but also selenoneine, a low molecular weight chemical form. I believe selenocysteine is a low molecular weight and is contained in the cytoplasm, so it is lost along with the water when it is removed from the fish meat during cooking. Please consider the chemical form of the selenium compound in your discussion of selenium residuals after cooking.
Reviewer 2 Report
The manuscript aims to report the retention factors of the Se of 10 species of fish consumed in Thailand, thus measuring the effect of two types of cooking on Selenium. The abstract and the introduction are clear and make you curious to read. Unfortunately, however, the article shows rather serious care in the presentation of the results.
Materials and methods: The formula of the Yiled Factor should be inserted in this section as well as its
definition, (currently in lines 183 and 184 of the results) considering that in the results and in the tables,
the Yield Factor is named.
Results: Selenium content (tables 3,4,5,6) must also be reported for 100 grams of product,
considering that the True Retention Factor is calculated starting from this data.
Furthermore, to relate the content of Se in the various species and in the various cooking methods,
it is necessary to refer to the concentration on the same quantity of product.
The same is true for the statistical analysis, some fish may have statistically significant Se contents
because they are found in a different content of the edible part.
In tables 3 and 4 the unit of measurement of the edible part is missing.
Tables 5 and 6 lack the unit of measurement for the Selenium content.
Check the caption of table 5: usually in the post hoc test results when the superscript letters are different
values show significantly different results: please check line 241.
Tables 3 -4 and 5, - 6 have values with different significant rounding of the digits after the comma,
in the values referred to the Se.
The values of the retention factors of some species do not coincide when comparing tables 3, 4 and 5, 6.
Discussion: The two-way anova allows to determine the effect of two variables and their interaction;
from the table and from the discussion of the results this data is not understood; example: lines 301-306 and 320-323.
What are the species of fish and the cooking that causes this? What data in the tables are you referring to?
Round 2
Reviewer 2 Report
The authors answered and corrected the errors, according to the indications.